# SuperFormer: Superpixel-based Transformers for Salient Object Detection

## Abstract

Images often have local redundant information that can strain the training of deep neural networks. An effective way to reduce spatial redundancy and image complexity is to over-segment with superpixels. With a fast, linear computational complexity, Simple Linear Iterative Clustering (SLIC) generates superpixels by grouping pixels as a function of colour similarity and spatial proximity. However, it is challenging and non-trivial to train a model on over-segmented images with dynamic graph structure and low spatial inductive bias. In order to train on unstructured data, graph neural networks (GNNs) can be applied to classify each superpixel for salient object detection (SOD) by considering a set of superpixels as graphs. Although other works on graph classification or node classification were able to utilize pre-defined edge information or GNNs, naive applications on superpixel graphs do not translate trivially. Our proposed **SuperFormer** method introduces new feature attributes for superpixels and a dynamic positional encoding for heterogeneous spatial graphs to achieve state-of-the-art results in salient object detection for low model complexity.

## 1 Introduction

Superpixel segmentation is the technique of grouping image pixels based on shared descriptive attributes like color and spatial similarity. The motivation behind this segmentation is to partition the image into meaningful regions, significantly reducing data complexity compared to individual pixels. Salient Object Detection (SOD) further refines this by classifying regions as foreground or background, a binary classification within semantic segmentation. Superpixel SOD combines both processes, classifying groups of superpixels, derived from the image, as either foreground or background.

Several pixel-wise SOD methods (Zhao et al., 2019b; Liu et al., 2018; 2021a) demand a substantial number of model parameters, primarily because of the encoder-decoder architecture (Hinton & Salakhutdinov, 2006) involving down-sampling and up-sampling. This architecture is essential for extracting semantic information. The encoder's role is to establish a broad receptive field through spatial reduction, while the decoder's function is to map low-resolution feature maps back to high-resolution predictions. Architectures that employ structures similar to UNet, (Zhao et al., 2019b; Liu et al., 2018; Ronneberger et al., 2015), which concatenate intermediate feature maps to the decoder require even more memory.

Over-segmenting an image reduces computational complexity and memory usage in downstream tasks by preserving boundaries and reducing degrees of freedom. However, extracting semantic information for image salient object detection (SOD) is challenging due to the unstructured nature of clustered segments, characterized by shape and location shifts. Recent methods, such as Suzuki et al. (2018a) and Yang et al. (2020), use a CNN for feature-level semantics before applying over-segmentation, but this still demands a substantial number of parameters and memory.

Other methods employ graph convolutional networks (GCNs) (Kipf & Welling, 2016; Veličković et al., 2017; Dadsetan et al., 2021; Avelar et al., 2020) for spatial aggregation, reducing computational requirements compared to pixel-level processing, but they do not incorporate structural and positional segment information. These methods perform well on simple datasets with low input variance, but struggle to generalize to more complex datasets with high input variance. Hence, learning effectively on over-segmented images remains a significant challenge.

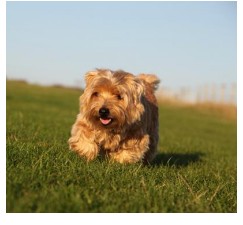 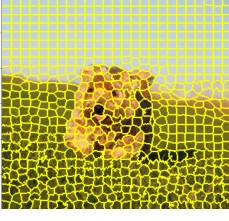 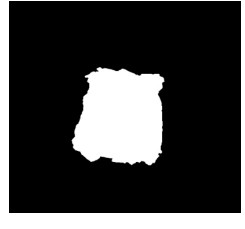 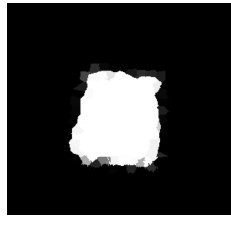

| (a) Raw Image | (b) Superpixel Segmentation | (c) Pixel-wise Saliency Mask | (d) Superpixel Saliency Mask |

Figure 1: **Problem formulation**. We oversegment a raw image (a) into superpixels (b), and use the segmentation map to transform the ground truth pixel-wise saliency mask (c) into superpixel saliency mask (d). Our problem formulation is expanded in Section 3.1.

We have observed that while superpixels reduce spatial degrees of freedom effectively, there is a need for representation methods and architectures to model such spatially heterogeneous representations. Given this observation, we propose **SuperFormer**, a superpixel-based Transformer model that can effectively learn on over-segmented images.

We evaluate our method on three SOD datasets: DUTS-TE (Wang et al., 2017), DUTS-O (Yang et al., 2013), and ECSSD (Yan et al., 2013). SuperFormer achieves SOTA F1-score results compared to other graph convolutional methods by 16.6%. Additionally, given similar model complexity, SuperFormer also achieves SOTA results compared to well-established pixel-wise methods by 5.8%.

Our contributions can be summarized as follows:
**(1)** We introduce a novel approach in the literature, utilizing Transformers to model superpixel representations. Our results show that abstract representations like superpixels can yield significantly improved outcomes with low model complexity.
**(2)** We incorporate shape attributes of superpixels using Fourier descriptors (Zahn & Roskies, 1972), enabling the representation of superpixel shape, size, and rotation.
**(3)** We introduce dynamic centroid positional embedding to tackle spatial heterogeneity in superpixel graphs. Rather than learning individual parameters for each superpixel index, we represent the spatial relationship using the Euclidean centroids of each superpixel as a function.

## 2 RELATED WORKS

### 2.1 PIXEL-WISE SALIENT OBJECT DETECTION

Pixel-wise SOD identifies regions of interest on an image, effectively classifying each pixel in the image as foreground or background. CNN-based approaches have been dominant in both RGB (Fan et al., 2020; Pang et al., 2020; Lee et al., 2022; Xie et al., 2022; Wu et al., 2022), and RGB-D (RGB + depth) (Chen & Li, 2018; 2019; Fu et al., 2020) pixel-wise SOD and achieved promising performance.

RGB SOD relies solely on a color image for object detection without a depth map. Many methods (Fan et al., 2020; Pang et al., 2020; Zhao et al., 2020) leverage an encoder-decoder structure with skip connections resembling a UNet-like structure (Ronneberger et al., 2015). Some works introduced the idea of fusing spatial and channel attention mechanism with CNN to discriminate the features even further (Fan et al., 2020; Chen et al., 2020; Piao et al., 2019) or pixel-wise contextual attention (Liu et al., 2018). Other works utilized recurrent networks to refine the saliency map sequentially (Zhang et al., 2018; Liu et al., 2019; Chen & Fu, 2020). In addition, some works approached the problem with multi-modal learning such as fixation prediction (Wang et al., 2018), image caption (Zhang et al., 2019), and edge detection (Zhao et al., 2019b; Ji et al., 2020; Zhang et al., 2020) to improve SOD accuracy.

For RGB-D SOD, the task assumes access to a depth mask aligned with the RGB image. Although depth features are not easily accessible, many works achieved significant improvements by utilizing such features. Some methods simply adopted feature fusion methods such as concatenation, summation, or multiplication (Chen & Li, 2019; Fu et al., 2020). Others generated spatial or channel attention as a function of depth cues to enhance the RGB features (Li et al., 2020a;b; Zhao et al.,

2019a). With the rapid increase of interest in Vision Transformers (Liu et al., 2021b; Chen et al., 2021; Dosovitskiy et al., 2020), some approached SOD from a sequence-to-sequence perspective and proposed a purely Transformer based SOD for both RGB and RGB-D (Liu et al., 2021a). In spite of all the success with pixel-wise SOD, these UNet-based methods require a significant number of trainable model parameters and corresponding computational effect, motivating the development in this paper of superpixel-based SOD.

## 2.2 SUPERPIXEL-BASED FEATURE EXTRACTION

He et al. (2015) treat the segmented superpixels as a 1D array, and extracts the contextual information from spatial kernels by calculating colour uniqueness. Colour distribution was a key factor that the authors determined to be a crucial factor in detecting saliency. Hence, two input sequences of colour uniqueness and colour distribution were trained with CNNs to detect salient objects.

To overcome the shortcomings of 1D representations, other methods proposed applying CNNs on 2D data either before or after superpixel segmentation (Zohourian et al., 2018b; Suzuki et al., 2018b). Zohourian et al. (2018b) proposed treating the initial grid generated by the SLIC algorithm as the topological basis in projecting the superpixel lattice. As a result, spatial inductive bias (Mitchell et al., 2017) is established for CNN operations. In addition, colour attributes, position features, and Local Binary Patterns (Ojala et al., 1994) are features used to describe each superpixel for model fitting.

Suzuki et al. (2018b) then proposed a hybrid model that applied naive CNNs before superpixel segmentation, and generalized convolution on generated superpixels. The generalized convolution aggregates based on an adjacency matrix that indicates which superpixels are neighbours. In addition, dilation is applied to the adjacency matrix to increase the receptive field of the kernels. Although the proposed method shows promising results, compared to pixel-wise naive CNN SOD, it still inherits the problem of heavy computation from both the early layers of CNN and generalized CNN.

Graph convolutional networks (Atwood & Towsley, 2016; Duvenaud et al., 2015; Hamilton et al., 2017) are spatial methods applied to learning representations of complex and unstructured data, in contrast to the grid-like structure of images. Since the number of vertices and edges of the graphs can vary arbitrarily, they become a powerful tool for data representation in irregular domains. Graph-based approaches utilize the notion of adjacency, defined by the edges in a graph, and perform graph convolution by aggregating features from the local neighbour nodes (Monti et al., 2017) instead of the entire graph. The non grid-like structure of superpixel images having an arbitrary number of superpixels make graph representation learning models a suitable candidate for superpixel images.

Monti et al. (2017) proposed the MoNET framework for general geometric data. The weighting of neighbourhood aggregation is determined through a learnable scaling factor based on geometric distances. Veličković et al. (2017) proposed Graph Attention Networks (GAT) that benefit from self-attention for weighting the neighbourhood aggregation. Avelar et al. (2020) then proposed joining superpixels and GAT to classify images. Although their proposed method performed worse than pixel-wise methods, their work opened an interesting branch of research on applying deep learning beyond regularly-gridded images. Dadsetan et al. (2021) similarly incorporated GCNs with superpixels to detect agricultural nutrient deficiency stress from aerial imagery. Instead of connecting nodes with their spatial neighbours, the nodes are fully connected so that information is propagated globally. The weighting of the edges is determined by colour similarity.

Extracting semantic features from unstructured superpixels is challenging as the process is non-trivially based on the constraints of standard CNNs. The shortcomings of GCNs on superpixels are that no shape or positional information is represented in the features for fully-connected graphs, and that the receptive field is low for GCNs that aggregate through pre-determined neighbours. In the next section, we propose SuperFormer to address the spatial heterogeneity of superpixels and to incorporate Fourier descriptors for shape representations.

## 3 METHODOLOGY: SUPERFORMER

In this section, we first describe the transformation of pixel-wise SOD into a superpixel SOD problem. Then, we introduce our superpixel representation and Transformer-based model.

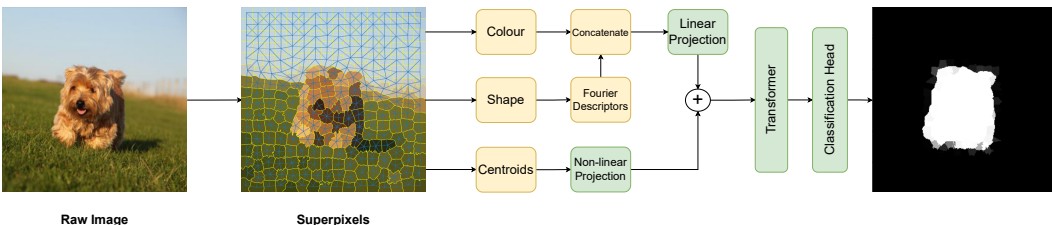

Figure 2: **SuperFormer overview.** A raw image is oversegmented into superpixels which contain low-dimensional abstract information about the image. Descriptive attributes of each superpixel are linearly projected and element-wise added to the non-linear projection of the centroids to inject spatial information. The resulting graph-like attributes are globally attended through Transformers and then passed through the final classification head for saliency detection.

## 3.1 PROBLEM FORMULATION

Given an image input $x \in \mathbb{R}^{H \times W \times 3}$ with height $H$, width $W$, and RGB channels, the corresponding pixel-wise label space is defined as $y \in \mathbb{R}^{H \times W}$, where each element in $y$ represents the saliency (foreground or background).

For a desired number $K$ of approximately equal-sized superpixels (number of segments) and compactness parameter $P$, the SLIC algorithm (Achanta et al., 2010) maps the raw pixels $x$ to superpixel index $x_l \in \mathbb{R}^{H \times W}$, where each spatial pixel is assigned to one of the superpixel indices $x_l \in \{1, 2, \ldots, K\}$ and $l$ is the number of image pixels.

Let $c^k \in \mathbb{R}^{n_k \times 2}$ be the Euclidean coordinates of the superpixel indices $x_l$ belonging to the $k^{th}$ superpixel. Additionally $z^k = x[c^k]$ assigns RGB values to the coordinates in set $c^k$. Then the corresponding superpixel label $y_s^k$ for $k \in 1, 2, \ldots, K$ can be obtained by averaging the pixel-wise mask values from a collection of raw pixel labels $c^k$ as follows:"

$$y_s^k = \texttt{Mean}(y[c^k]) \tag{1}$$

Therefore, the superpixel SOD task is to model a Transformer-based function $f$ as a function of superpixel representation function $g$ and positional encoding function $v$ as follows:

$$\mathcal{Z} = \{z^1, z^2, \ldots, z^K\} \tag{2}$$

$$\mathcal{C} = \{c^1, c^2, \ldots, c^K\} \tag{3}$$

$$\hat{y}_s^k = f\big(g(x_l, \mathcal{Z}), v(\mathcal{C})\big) \tag{4}$$

Finally, superpixel predictions are mapped back to regular pixel space to be compared against SOTA pixel-wise SOD methods:

$$\hat{y}[c^k] = \hat{y}_s^k \tag{5}$$

Now we define our superpixel representation function $g$ and our Transformer-based model (Super-Former) $f$ in the next two subsections.

## 3.2 SUPERPIXEL REPRESENTATION

Our observation highlights local redundant information in images, necessitating the grouping of similar pixels to reduce redundancy while preserving relevant details. Superpixel segmentation serves this purpose by significantly reducing both input representation and model complexity. This transformation from pixel grids to graph-like structures introduces challenges in extracting semantic information through convolutional kernels and filters. The graph-based input requires additional visual cues for salient object detection.

Color distribution serves as a fundamental cue for saliency detection. However, representing variable-sized superpixels in a tensor form suitable for optimization algorithms like stochastic gradient descent poses challenges. Therefore, a statistical representation of each superpixel becomes essential to abstractly capture the color distribution.

We will employ the most basic distribution for now, while keeping more advanced options as potential avenues for future research. Specifically, a 6-D vector representing the mean and standard deviation

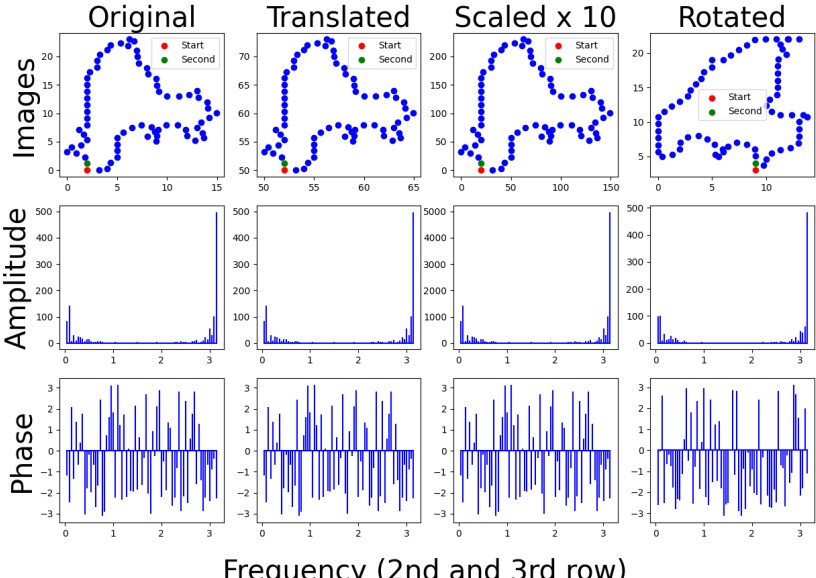

Figure 3: **Fourier descriptors**. Given a sample superpixel contour (left top), the amplitude and phase of the Fourier Transform is visualized for translation (2nd column), scale (3rd column), and rotation (4th column). Observe that for there is no change for translation, only magnitude changes for scaling, and changes in phase for rotation. The "Start" and "second" legend represents the starting point and the direction of the complex sequence, respectively.

(STD) of each RGB channels will be utilized. Given the raw pixel RGB intensities $x_{\{R,G,B\}}$, then $g_{\text{colour}}(x[c^i])$ with mean $\texttt{M}$ and standard deviation $\texttt{SD}$ is defined as:

$$g_{\text{colour}}(z^k) = \left[ \texttt{M}\left(x_R[c^k]\right), \texttt{M}\left(x_G[c^k]\right), \texttt{M}\left(x_B[c^k]\right), \texttt{SD}\left(x_R[c^k]\right), \texttt{SD}\left(x_G[c^k]\right), \texttt{SD}\left(x_B[c^k]\right) \right] \quad (6)$$

In addition to colour, shape is an important aspect in determining the saliency of a superpixel. To represent shape in vector form, we sample the contour (`Contour`) of each superpixel and then use the Fourier transform to obtain the complex coefficients of amplitude and phase that implicitly define the shape of each superpixel.

A set of Euclidean coordinates $u^k = \{(x_1^k, y_1^k), (x_2^k, y_2^k), \ldots, (x_r^k, y_r^k)\}$, defining the boundary of a superpixel, is transformed into the complex plane (`Complex`) $b^k = \{x_1^k + iy_1^k, x_2^k + iy_2^k, \ldots, x_r^k + iy_r^k\}$:

$$U = \texttt{Contour}(x_l) = [u^1, u^2, \ldots, u^K], B = \texttt{Complex}(U) \quad (7)$$

The resulting coefficients of the Fourier transform $\texttt{FT}(B)$ have translation invariance (ignoring the zero frequency coefficient) while the size of the superpixel is reflected in the amplitude and the rotation of the superpixel is reflected in the phase.

Formally, shape representation for a superpixel is defined as:

$$g_{\text{shape}}(B) = \texttt{FT}(B) = [\text{amp}_1, \text{amp}_2, \ldots, \text{amp}_{r-1}, \text{phase}_1, \text{phase}_2, \ldots, \text{phase}_{r-1}] \quad (8)$$

Then, $g_{\text{colour}}$ and $g_{\text{shape}}$ are concatenated and linearly projected with $W_a \in \mathbb{R}^{(2(r-1)+6) \times d}$ to be the input for the Transformer-based model in Section 3.3:

$$g(x_l, \mathcal{Z}) = \texttt{Concat}\left(g_{\text{shape}}(\texttt{Complex}(\texttt{Contour}(x_l))), g_{\text{colour}}(\mathcal{Z})\right) W_a \quad (9)$$

### 3.3 SUPERFORMER

CNNs assume a certain level of spatial inductive bias (Mitchell et al., 2017) where it expects the input to have high spatial consistency, such as a grid-like pixel images. Since superpixels display a graph-like structure that is dependent on the location and the size of each superpixel, CNNs are not a good fit for the task. Therefore, an architecture in the form of graph neural network is needed to model the superpixel representation. In this subsection, we will expand on the details of the Transformer-based architecture that we call **SuperFormer**.

Current SOTA learnable positional embeddings (Devlin et al., 2018) either assume a homogeneous positional relationship, where the index of the tokens are consistent within the spatial/sequential domain, or create a dictionary of learnable parameters to consider all possible relative positions (Huang et al., 2018). However, since the locations of superpixels display spatial heterogeneity, and creating learnable parameters for all possible 2D relative position is computational impractical, there is a need to express positional information as function of the location of the superpixels.

To address this problem, we propose a new method called Dynamic Centroid Positional Embedding (DCPE) that non-linearly projects a positional encoding determined by centroids of each superpixel.

Given the centroid Euclidean coordinates of each superpixel $c^i = \{(x_1^i, y_1^i), (x_2^i, y_2^i), \ldots, (x_{n^i}^i, y_{n^i}^i)\}$, positional information $p \in \mathbb{R}^{K \times 2}$ is represented as the centroids.

Then, learnable parameters $W_{p,1} \in \mathbb{R}^{2 \times 64}$ and $W_{p,2} \in \mathbb{R}^{64 \times d}$ can be used to non-linearly (ReLU activation) project the centroids to a $d$ dimensional vector, which addresses the problem of hetereogeneity of superpixel centroids. Formally, we have

$$v(p) = \text{ReLU}(pW_{p,1})W_{p,2} \tag{10}$$

The linearly projected descriptive attributes from equation 9 are added element-wise to the positional encoding equation 10 and then passed through the vanilla Transformer (`Transformer` (Vaswani et al., 2017)) for global self-attention applying a classification head consisting of one linear layer $W_C \in \mathbb{R}^{d \times 1}$ to predict the saliency of each superpixel:

$$f(g(x_l, \mathcal{Z}), v(\mathcal{C})) = \text{Transformer}\big(g(x_l, \mathcal{Z}) + v(h(\mathcal{C}))\big)W_C \tag{11}$$

In conclusion, with superpixel segmentation $x_l = \texttt{SLIC}(x)$ and a set of superpixel centroids $\mathcal{C} = \{c_1, c_2, \ldots, c_K\}$ the overall model $f$ is formulated as follows:

$$\hat{y}_s = f(g(x_l, \mathcal{Z}), v(h(\mathcal{C}))) \tag{12}$$

Binary cross-entropy is used as the objective function to minimize the loss between predicted superpixel saliency $\hat{y}_s$ and the ground-truth superpixel saliency $y_s$.

## 4 EXPERIMENTS

In this work, we have two primary objectives: firstly, to illustrate a substantial performance gap in the state-of-the-art methods related to superpixel-based approaches, and secondly, to establish that SuperFormer outperforms pixel-wise methods of comparable model complexity. Details of hyperparameter settings used in our experiments are shown in Table 1.

Table 1: Detailed hyperparameter settings of the experiments.

| Parameters | Settings | Parameters | Settings | Parameters | Settings |
|---|---|---|---|---|---|
| $P$ | 10 | # Points | 70 | Epochs | 400 |
| $K$ | 625 | Opt. | AdamW | Random-Flip | Yes |
| # TFM heads | 8 | LR | 0.001 | Resize-Crop | 224 |
| # Hidden dim | 16 | LR decay | 0.1 | Resolution | $224 \times 224$ |
| # TFM layers | 6 | LR patience | 50 | Pre-trained | False |

### 4.1 DATASETS

DUTS-Train (Wang et al., 2017) is chosen as the training set and tested on DUTS-TE (Wang et al., 2017), ECSSD (Yan et al., 2013), and DUTS-O (Yang et al., 2013) following the work developed in (Zhao et al., 2019b; Liu et al., 2018; 2021a; Lee et al., 2022). Currently, DUTS stands as the largest SOD dataset with predefined training and test splits. This dataset comprises 10,553 training images and 5,019 test images, all meticulously annotated by 50 individuals. It encompasses images from the ImageNet DET training and validation sets, offering a broad spectrum of foreground and background variability.

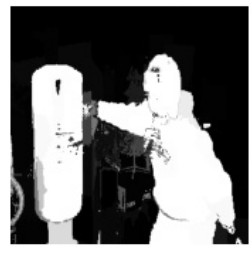 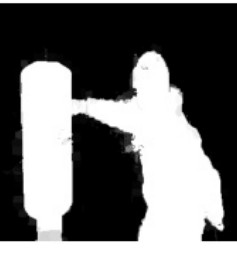 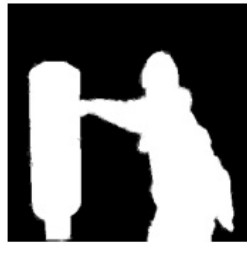 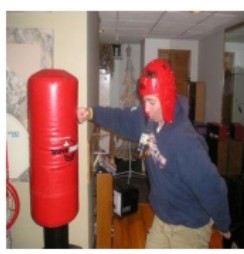

(a) 100 segments     (b) 1089 segments     (c) 10000 segments     (d) Raw Image

Figure 4: **Qualitative results on superpixel ground truth mask accuracy as a function of segment numbers.** Note that as the number of segments increases, the boundaries of the mask are less noisy since the superpixels are smaller and are able to locally segment finer details. However, even with a small number of segments, the visual quality of the superpixel gound truth hold up well.

Table 2: **Main results.** Table presents the evaluation results of SuperFormer (SF) on three well-known SOD datasets detailed in Section 4.1. It also includes a comparison with three state-of-the-art superpixel-based methods (GAT, SC, GF) and three state-of-the-art pixel-wise methods (VST, MobileUNetV2, TRACER). According to the results, SF boosts the F1-score by 16.6% on average compared to superpixel-based methods and by 5.8% on average compared to SOTA pixel-wise methods with similar model complexity. This includes SLIC and Fourier Descriptors processing time in the inference.

| Method | Model Mem. (MB) | Params. (M) | MACs (M) | Infer. (ms) | DUTS-TE MAE↓ | DUTS-TE maxF↑ | DUTS-O MAE↓ | DUTS-O maxF↑ | ECSSD MAE↓ | ECSSD maxF↑ |
|---|---|---|---|---|---|---|---|---|---|---|
| GAT | 0.42 | 1.59 | 2191 | 11 | 0.176 | 0.607 | 0.168 | 0.587 | 0.165 | 0.735 |
| SC | 6.08 | 1.50 | 950 | 2 | 0.142 | 0.655 | 0.130 | 0.628 | 0.122 | 0.784 |
| GF | 2.46 | 0.62 | 990 | 73 | 0.187 | 0.621 | 0.179 | 0.581 | 0.172 | 0.715 |
| VST | 178.24 | 44.48 | 23180 | 8 | 0.130 | 0.638 | 0.132 | 0.627 | 0.108 | 0.814 |
| MobileUNetV2 | 5.07 | 1.15 | 2058 | 2 | 0.104 | 0.658 | 0.105 | 0.642 | 0.096 | 0.816 |
| TRACER | 17.44 | 7.45 | 2130 | 25 | 0.094 | 0.714 | **0.090** | 0.718 | 0.083 | 0.857 |
| **SF (Ours)** | 2.46 | 0.61 | 990 | 2 | **0.092** | **0.765** | **0.090** | **0.757** | **0.077** | **0.881** |

## 4.2 METRICS

We adopt two widely-used evaluation metrics to evaluate our model performance. Specifically, maximum F1-score jointly considers precision and recall under the optimal threshold with a weight of 0.3 on precision. Mean Absolute Error computes pixel-wise average absolute error. To evaluate the model complexity, we also report the model memory and the number of parameters.

## 5 RESULTS

Table 2 shows the F1-score achieved by SuperFormer on the DUTS-TE, DUTS-O, and ECSSD datasets, as compared to related works: popular graph convolution method Graph Attention Network (GAT) (Veličković et al., 2017), SOTA superpixel convolution method Superpixel Convolution (SC) (Zohourian et al., 2018a), generalized Transformers for graphs (GF) (Dwivedi & Bresson, 2020), vision Transformer based method Visual Saliency Transformer (VST) (Liu et al., 2021a), SOTA light-weight model MobileUNetV2 (Sandler et al., 2018), and SOTA pixel-wise SOD model TRACER (Lee et al., 2022).

Considering models with similar complexity, SuperFormer consistently outperforms state-of-the-art graph convolution methods by 16.6% and pixel-wise saliency object detection methods by 5.8% in average F1-score across all three datasets . None of the compared methods, including GNN-based ones (GAT, GF), superpixel convolution (SC), and ours (SF), use pre-trained ImageNet weights. For a fair comparison, we trained VST, MobileUNetV2, and TRACER without ImageNet pre-training. Further discussion about not using ImageNet pre-training can be found in A.1

Figure 5 displays examples of the predicted saliency map generated by the six baseline models and SuperFormer. Observe that graph-based (superpixel-based) methods do a poor job with high levels of noise in the predictions, while pixel-wise methods do not capture the boundaries of objects well due to the low model complexity. However, even with low model complexity, SuperFormer is able

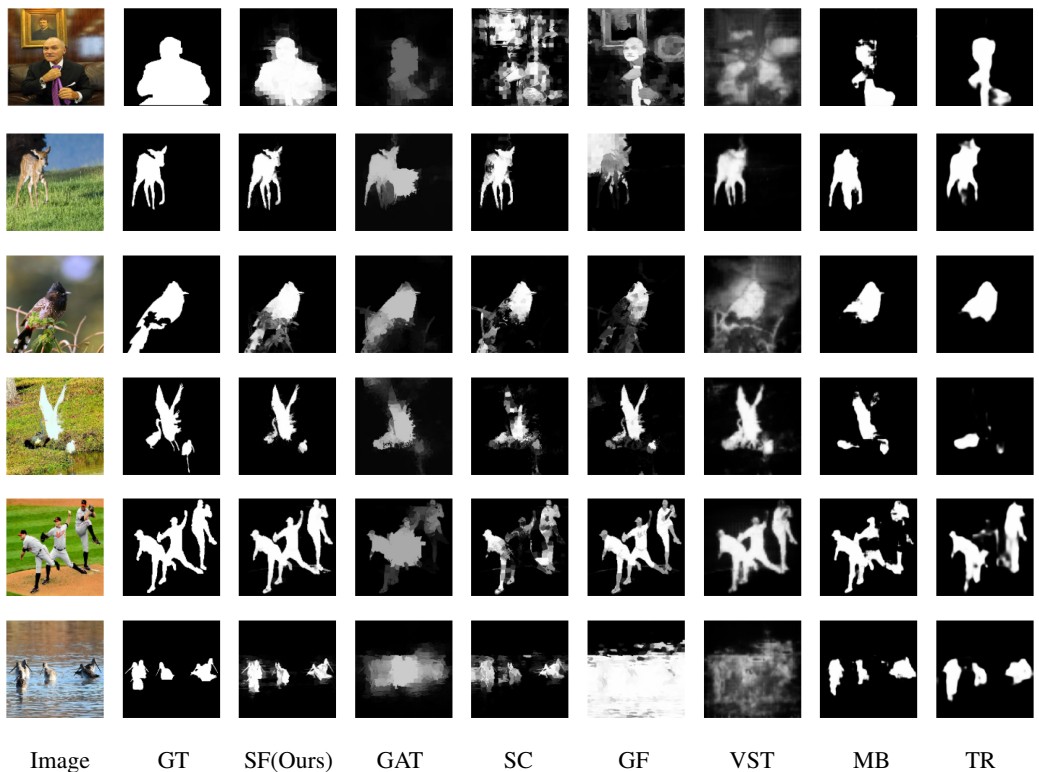

|  Image | GT | SF(Ours) | GAT | SC | GF | VST | MB | TR |

Figure 5: **Qualitative saliency map comparison.** Notice that the boundary of our predicted saliency map seamlessly aligns with the ground truth boundary, even with lower model complexity compared to other pixel-wise methods, namely VST (Liu et al., 2021a), MB (MobileUNetV2) (Sandler et al., 2018), and TR (TRACER) (Lee et al., 2022). Here, GT represents the ground truth.

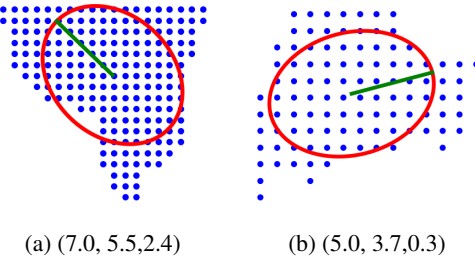

(a) (7.0, 5.5,2.4)                (b) (5.0, 3.7,0.3)

Figure 6: **Eccentricity Visualization.** Eccentricity can describe superpixel shapes in low-order representation by fitting ellipses, providing major axis length, minor axis length, and orientation. However, it offers less detail compared to Fourier descriptors. Blue dots represent pixels forming a superpixel, while (A, B, C) denote (major axis length, minor axis length, orientation in radians).

Table 3: **Impact of $K$.** As the segment count ($K$) grows, MAE decreases, and maxF increases. Note that while the number of parameters (model complexity) remains constant, the self-attention module in the Transformer block (Eq. equation 11) exhibits quadratic computation complexity with respect to the segment count ($K \times K$).

| K | MAE↓ | MAE△ | maxF↑ | maxF△ |
|---|---|---|---|---|
| 100 | 0.123 | – | 0.735 | – |
| 225 | 0.109 | 0.014 | 0.751 | 0.016 |
| 400 | 0.099 | 0.024 | 0.758 | 0.023 |
| 625 | 0.098 | 0.025 | 0.764 | 0.029 |

to understand the scene with such abstract representation and predict much improved salient maps compared to all six baselines.

Table 4: **SuperFormer Components.** Our baseline model with basic Transformers and learnable index positional embedding, we assess the impact of Fourier descriptors and dynamic centroid positional embedding on the DUTS-TE dataset. Notably, low-order representation (Eccentricity) yields marginal improvements over the baseline. However, the addition of dynamic centroid positional embedding (DCPE) and Fourier descriptors (FD) results in F1-score improvements of 0.015 and 0.036, respectively.

| Method | MAE$\downarrow$ | MAE$\Delta$ | maxF$\uparrow$ | maxF$\Delta$ |
|---|---|---|---|---|
| Baseline | 0.113 | – | 0.728 | – |
| + Ecc (Zhang & Lu, 2004) | 0.124 | -0.011 | 0.629 | -0.101 |
| + DCPE equation 10 - Ecc (Zhang & Lu, 2004) | 0.109 | 0.004 | 0.743 | 0.015 |
| + FD equation 8 | 0.098 | 0.015 | 0.764 | 0.036 |

In Table 4, we measure the isolated impact of Fourier descriptors and DCPE by evaluating a baseline model on DUTS-TE which consists of the colour attributes equation 6, and learnable index positional embeddings (Devlin et al., 2018). Then, we sequentially add DCPE equation 10 and Fourier descriptors (FD) 8 which improves the baseline F1-score by 0.015 and 0.036 respectively. We also study the impact of low order shape representation by replacing FD with Eccentricity (Zhang & Lu, 2004). Although Eccentricity can provide low dimensional information about the rough shape of superpixels, it does not provide enough features for the model to characterize the shapes. Eccentricity is visualized through Figure 6

## 5.1 ABLATION STUDIES

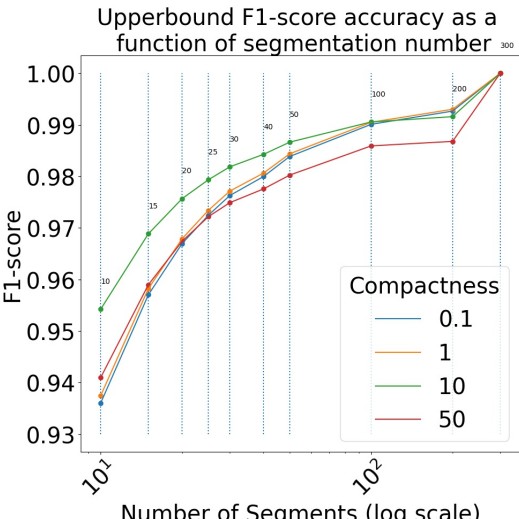

Figure 7: **Upperbound F1-score accuracy for superpixel segments.** We empirically examine the relationship between the number of segments ($K$) and the maximum achievable F1-score. Even with a relatively low number of segments (around 200 to 300), we maintain a high upper-bound F1-score accuracy of approximately 0.96.

We perform ablation studies on SuperFormer, analyzing the impact of various components and hyperparameters. Figure 7 shows how the weighted F1-score accuracy of the superpixel ground-truth map changes with the number of segments ($K$). This quantifies the loss, indicating the disagreement between the superpixel ground-truth map and the pixel-wise ground truth map.

Even with a low number of segments (300 or more), the maximum F1-score loss remains at only 0.04, regardless of the compactness ($P$). The influence of $K$ on the overall model accuracy is summarized in Table 3. You can also see the qualitative impact of the number of segments in Figure 4.

## 6 CONCLUSION

Our study tackles the integration of superpixel-based image segmentation into deep neural networks effectively. We propose innovative techniques like Fourier descriptors for superpixel shapes and dynamic centroid positional embedding for diverse superpixel coordinates. Despite their abstract nature and low model complexity, our approach excels in salient object detection across datasets.

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
