# OpenReview forum: "SuperFormer: Superpixel-based Transformers for Salient Object Detection"
_ICLR.cc/2024/Conference — ICLR 2024 Conference Withdrawn Submission_

### Official Review · Reviewer_tMkj · 2023-10-31

**Soundness:** 1 poor
**Presentation:** 2 fair
**Contribution:** 1 poor
**Rating:** 3
**Confidence:** 5

**Summary:**

This paper presents a superpixel-based salient object detection (SOD) method. Firstly, they transform the pixel-wise SOD into superpixel SOD by transforming the pixel-wise mask into superpixel masks. Additionally, they propose a transformer-based framework including a dynamic centroid positional embedding (DCPE).

**Strengths:**

- Utilizing superpixel for salient object detection is interesting.
- Provide technical details in Section 3, so that the readers could reproduce the methods based on descriptions.

**Weaknesses:**

- Though Section 3 provides technical details, the motivation is somehow unclear or from technical aspects, which may lack novelty.
- The experiments are not fully conducted and the "new state-of-the-art" performance stated in the paper is doubtful. In Table 2, only 6 methods are compared and most of them are proposed before 2020.
- The writing style needs to be improved. Inappropriate words and abbreviations should be avoided in formal academic writing.
- The supplementary materials are missing.

**Questions:**

- Please compare your method with more superpixel-based methods, light weight pixel-wise methods, and salient object detection methods that are proposed in recent years (2021, 2022, 2023)?

---

### Official Review · Reviewer_DeSV · 2023-11-01

**Soundness:** 2 fair
**Presentation:** 3 good
**Contribution:** 2 fair
**Rating:** 3
**Confidence:** 4

**Summary:**

This paper centers its attention on superpixel segmentation for the SOD task. It endeavors to leverage the Transformer for processing superpixel representations, employ Fourier transformation to depict the shape attributes of superpixels, and apply dynamic centroid positional embedding to address the issue of superpixel heterogeneity. With SOTA models, it appears to attain commendable performance.

**Strengths:**

1.	Small number of model parameters for easy deployment.
2.	The structure of paper is well organized.

**Weaknesses:**

1.	The paper's justification for the need for superpixels is not persuasive and there is limited novelty to use Fourier descriptors for extraction of feature.
2.	Incorporating the non-differentiable SLIC method can present obstacles when attempting end-to-end training or integration into other deep learning models.
3.	The experimental results are quite unusual, with the SOTA algorithm differing significantly from the results presented in the original paper (VST, etc.).
4.	Pixel-level sod models are insufficient and the visualization results are hardly satisfying.
5.	Details of the evaluation on speed and computation are missing.

**Questions:**

1.	Why not use the saliency maps or trained weights released in the SOTA paper when comparing other algorithms? The data in the paper is far from the real SOTA.
2.	The correspondence between the amount of computation and the speed of inference is strange. Please describe the hardware environment in which the evaluation was performed. Also explain the main difference with GF that leads to a comparable number of parameters and computation but a large difference in inference speed.

---

### Official Review · Reviewer_P45W · 2023-11-02

**Soundness:** 2 fair
**Presentation:** 2 fair
**Contribution:** 2 fair
**Rating:** 3
**Confidence:** 4

**Summary:**

This paper advocates employing SLIC-generated superpixels as a foundation. It suggests utilizing GNN to address irregular shapes, employing Fourier transform for shape characterization, and incorporating dynamic centroid positional embedding to manage non-uniform superpixel positions. The resulting SuperFormer demonstrates state-of-the-art performance in Salient Object Detection (SOD).

**Strengths:**

1. This approach presents a simple yet effective technique for extracting features from superpixels.
1. The proposed method is fast compared with other SOD methods.

**Weaknesses:**

1. The novelty is constrained, as the integration of superpixels with Transformers has been explored in prior works [1][2].
2. The reported performance of the baseline is inaccurate. The VST model's MAE and F-measure on the DUTS-TE dataset are stated as 0.037 and 0.877, respectively.
3. There is a lack of information regarding the specific architecture employed in this paper, which makes the comparison in Table 2 not clear enough.

[1] Vision Transformer with Super Token Sampling

[2] Superpixel Transformers for Efficient Semantic Segmentation

**Questions:**

Please see the weaknesses.